# The Concurrent Application of Phosphogypsum and Modified Biochar as Soil Amendments Influence Sandy Soil Quality and Wheat Productivity

**DOI:** 10.3390/plants13111492

**Published:** 2024-05-28

**Authors:** Mohssen Elbagory, Eman M. Shaker, Sahar El-Nahrawy, Alaa El-Dein Omara, Tamer H. Khalifa

**Affiliations:** 1Department of Biology, Faculty of Science and Arts, King Khalid University, Mohail 61321, Assir, Saudi Arabia; mhmohammad@kku.edu.sa; 2Soil Improvement and Conservation Research Department, Soils, Water, and Environment Research Institute (SWERI), Agriculture Research Center (ARC), Giza 12112, Egypt; e_sh_alla@yahoo.com; 3Soil Microbiology Research Department, Soils, Water, and Environment Research Institute (SWERI), Agriculture Research Center (ARC), Giza 12112, Egypt; sahar.elnahrawy@yahoo.com

**Keywords:** phosphogypsum, cotton stalk biochar, rice straw biochar, modified biochar, loamy sand soil, wheat yield

## Abstract

Sandy soil covers a significant portion of Egypt’s total land area, representing a crucial agricultural resource for future food security and economic growth. This research adopts the hypothesis of maximizing the utilization of secondary products for soil improvement to reduce ecosystem pollution. The study focuses on assessing the impact of combining phosphogypsum and modified biochar as environmentally friendly soil amendments on loamy sand soil quality parameters such as soil organic carbon, cation exchange capacity, nutrient levels, and wheat yield. The treatments were T_1_: the recommended NPK fertilizer (control); T_2_: 2.5 kg phosphogypsum m^−2^ soil; T_3_: 2.5 kg rice straw biochar m^−2^ soil; T_4_: 2.5 kg cotton stalk biochar m^−2^ soil; T_5_: 2.5 kg rice-straw-modified biochar m^−2^ soil; T_6_: 2.5 kg cotton-stalk-modified biochar m^−2^ soil; and T_7_ to T_10_: mixed phosphogypsum and biochar treatments. The results revealed that the combined use of phosphogypsum and modified cotton stalk biochar (T_10_) significantly enhanced soil organic carbon (SOC) by 73.66% and 99.46% in both seasons, the soil available N both seasons by 130.12 and 161.45%, the available P by 89.49% and 102.02%, and the available K by 39.84 and 70.45% when compared to the control treatment. Additionally, this treatment led to the highest grain yield of wheat (2.72 and 2.92 Mg ha^−1^), along with a significant increase in straw yield (52.69% and 59.32%) compared to the control treatment. Overall, the findings suggest that the combined use of phosphogypsum and modified biochar, particularly cotton-stalk biochar, holds promise for improving loamy sand-soil quality and wheat productivity.

## 1. Introduction

In line with the 2030 Sustainable Development Goals (SDGs) of the United Nations, improving soil fertility has become crucial for maintaining ideal soil qualities, sequestering carbon, and providing plants with enough nutrients in a balanced manner. These efforts are essential for ensuring soil security, sustaining high crop yields, and bolstering the rural economy [1]. Sandy soil, covering a staggering 96% of Egypt’s total land area, stands out as the nation’s most significant agricultural resource [2]. Its sustainable management holds the key to meeting food demand, fostering economic growth, and ensuring national security.

The intensification of agricultural practices and associated manufacturing processes has led to a substantial generation of agro-industrial waste. This increase in production has exerted significant pressure on the environment, adversely affecting agro-system resources [3]. The production of phosphoric acid from rock phosphate in the phosphate fertilizer industry, which results in the generation of phosphogypsum as a byproduct, illustrates this challenge. Despite an annual production of 160 million tons worldwide, only 15% of this amount is utilized [4]. In Egypt alone, approximately 11–14 million tons of phosphogypsum are generated annually from phosphoric acid production [5,6]. This waste material contains heavy metals and fluorine, posing a serious risk of pollution to the atmosphere [7,8] and soil organism communities [9]. Indeed, reusing phosphogypsum in sandy soil could present a sustainable solution for managing accumulated quantities of it. By incorporating phosphogypsum into sandy soil, we not only address its disposal but also enhance soil fertility and potentially boost crop yields. This approach aligns with sustainability principles by repurposing a byproduct in a beneficial manner while minimizing waste and environmental impact.

Phosphogypsum possesses physical and chemical characteristics that make it suitable for use as a soil amendment or agricultural fertilizer [10,11,12,13,14]. Studies have indicated that incorporating phosphogypsum into soil does not alter its acidity but enhances nutrient solubility [15,16], thereby facilitating deeper root penetration [17,18]. However, research on its impact on sandy soil remains limited. Some studies, such as that by Karbout et al. [19], have demonstrated that phosphogypsum application to sandy soil can significantly affect soil structure, nutrient availability, and erosion resistance. Additionally, findings by Bossolani et al. [20] support the notion that phosphogypsum improves soil fertility and increases plant macronutrient uptake. Mahmoud et al. [21] have also shown that phosphogypsum application has significant effects on organic matter content and soil nutrient availability.

Because soil carbon storage is essential to many different biogeochemical processes occurring in the soil, it is a critical indicator of soil fertility and health [22]. Carbon sequestration in soils presents a viable approach to offset increased CO_2_ efflux from soil [23]. Biochar, a multifunctional carbon material, holds promise as a solution for diverse environmental and agricultural challenges, and it is extensively utilized as a soil modifier to enhance soil properties and productivity [24,25,26]. Moreover, previous studies have demonstrated that adding biochar to sandy soil can augment soil organic carbon content, nutrient availability, and crop production [27,28,29,30,31,32,33,34,35,36,37]. However, negative effects on crop yield have been observed at high application rates. This decrease in yield is attributed to elevated pH levels, leading to nitrogen immobilization and reduced its to plants [38], as well as the inhibition of microbial communities due to certain reactive compounds and heavy metals present in biochar [39]. One of the primary challenges facing biochar’s efficacy stems from the diverse feedstocks used in its production. These feedstocks, primarily agricultural residues, undergo combustion at temperatures ranging from 300 to 1000 °C with limited or no oxygen. Each feedstock possesses unique qualities that directly impact the characteristics and performance of the resulting biochar [40,41,42].

Consequently, the chemical treatment or modification of biochar with acids is employed to create biochar with specialized functional groups, enhanced adsorption capacity, and acidity [43,44,45,46]. During the modification process with H_2_SO_4_, there is a notable increase in carboxyl functional groups, which, owing to their acidic nature and proton-exchange capability, play a crucial role in soil resistance to acidification [47]. Acid treatments, whether administered prior to or following pyrolysis, serve multiple purposes, such as augmenting surface area, reducing pH levels, and eliminating impurities and metallic precipitates. These processes collectively enhance the cation sorption capacity of biochar [48,49]. The application of modified biochar to soil has been shown to improve nutrient availability and crop yield [50,51,52,53].

This study looked at the effects of co-applying modified biochar and phosphogypsum on soil organic carbon levels, fertility, and wheat yield in sandy loam soil.

## 2. Materials and Methods

### 2.1. Experimental Location and Design

The lysimeter experiment took place at the Soil Improvement and Conservation Greenhouse of the Sakha Agricultural Research Station in Kafr El-Sheikh Governorate, Egypt (31°5′38.21″ N and 30°56′54.92″ E), during the growing seasons of 2021–2022 and 2022–2023. The aim was to investigate the impact of the co-application of phosphogypsum and modified biochar on enriching soil organic carbon, improving the fertility of sandy soil, and enhancing wheat yield.

Thirty lysimeters were utilized, each with dimensions of 2 m^2^ in width and 1 m in depth. Table 1 succinctly summarizes the soil characteristics and nutrient levels.

The experimental design followed a randomized block design with three replications. The experiment included the following treatments:

T_1_: Control (recommended NPK fertilizer)

T_2_: Phosphogypsum application at a rate of 2.5 kg m^−2^ soil

T_3_: Rice straw biochar application at a rate of 2.5 kg m^−2^ soil

T_4_: Cotton stalk biochar application at a rate of 2.5 kg m^−2^ soil

T_5_: Rice-straw-modified biochar application at a rate of 2.5 kg m^−2^ soil

T_6_: Cotton-stalk-modified biochar application at a rate of 2.5 kg m^−2^ soil

T_7_: Mixed phosphogypsum with rice straw biochar application

T_8_: Mixed phosphogypsum with cotton stalks biochar application

T_9_: Mixed phosphogypsum with rice-straw-modified biochar application

T_10_: Mixed phosphogypsum with cotton-stalk-modified biochar application

Each treatment was applied to the lysimeters according to the experimental design, and relevant parameters were monitored throughout the duration of the study.

### 2.2. Materials

After collection, phosphogypsum (PG) was obtained from a fertilizer industry factory located in Abu-Zaable district, El-Sharkia Governorate, Egypt. PG is an acidic waste material with the following composition: pH: 3.2, CaO: 35.9%, SO_3_ (44.08%), SiO_2_: 9.95%, P_2_O_5_: 7.38%, Fe_2_O_3_: 1.64%, and traces of Na_2_O, TiO_2_, and F.

Rice straw and cotton-stalk biochar were prepared according to the method outlined by Mosa et al. [54]. The process involves drying the raw materials at 70 °C until a constant weight is achieved, followed by pyrolysis at 550 °C for 2 h under oxygen-limited conditions in a muffle furnace. The resulting biochar is then ground. Modified biochar was prepared by shaking 1 kg of rice straw and cotton stalk biochar with 1 L of sulfuric acid (0.1 M) at 150 rpm for 4 h. The mixture was shaken, then filtered, cleaned with distilled water, and allowed to dry for 24 h at 70 °C. Chemical analysis was conducted according to [55,56]. Table 2 displays the chemical parameters of the raw materials, biochar, and its modified form.

### 2.3. Agricultural Practices

During the control treatment, a single dosage of super-phosphate (15.5% P_2_O_5_) phosphorus fertilizer was administered at a rate of 476 kg ha^−1^. During lysimeter preparation, phosphogypsum, biochar, and super-phosphate were mixed into the soil’s top 0–20 cm. On 16 November 2022, wheat grains (*Triticum aestivum* L., variety Sakha 95) were seeded at a rate of 144 kg ha^−1^. Two equal splits of urea (46% N), a nitrogen fertilizer, were applied at a rate of 90 kg ha^−1^ 25 and 50 days after planting. Additionally, during the second irrigation, 119 kg ha^−1^ of potassium sulfate (48% K_2_O) was added. Chemical fertilizers were applied in all treatments except for those treated with phosphogypsum, where phosphorus fertilizers were omitted.

### 2.4. Soil and Plant Analysis

In order to investigate soil properties, surface soil samples (0–20 cm depth) were gathered prior to planting and following wheat harvesting in both seasons. These samples were analyzed using the techniques described by Pansu et al. [57] and Carter and Gregorich [58].

A one-meter-square section of wheat plants was sampled at the maturity stage in order to assess the characteristics of the plants. Grain and straw yield in kilograms per plot were used to calculate plant biomass, which was then converted to megagrams per hectare (Mg ha^−1^).

During the experiment, six readings were initially taken. Subsequently, only three replicates exhibiting closely aligned results were utilized.

### 2.5. Statistical Analyses

The study utilized R software (version 4.3.1) to rigorously analyze and visualize the obtained results [59]. Initially, one-way ANOVA was applied to evaluate the variance between treatments, followed by Tukey’s multiple comparison tests for detailed assessments. Bar plots were then generated to visually represent the results of ANOVA, accompanied by standard error bars to emphasize any significant differences between treatments. To delve into the complex relationships among various soil parameters, wheat yield, and treatments, the Factoextra package was employed [60]. This package facilitated the visualization of these relationships, allowing for a comprehensive understanding of how the different treatments impact soil health and crop productivity.

## 3. Results

### 3.1. Soil Quality

According to Table 3, the ANOVA results reveal the significant effects of phosphogypsum and biochar treatments on soil organic carbon (SOC), cation exchange capacity (CEC), and the availability of soil nutrients (N, P, and K) in both seasons.

Compared to the control treatment, T_10_ exhibited higher SOC levels, resulting in an increase of 73.66% and 99.46% in both seasons, respectively (Figure 1). Furthermore, the modified biochar treatments (T_5_ and T_6_) had a more pronounced positive effect on SOC compared to the normal biochar treatments (T_3_ and T_4_). No statistically significant differences in SOC were observed between the co-application of phosphogypsum and biochar treatments.

In the 2021/2022 season, no significant differences in CEC were observed due to biochar treatments or co-application treatments (Figure 2a). However, in the second season, the co-application of phosphogypsum and normal biochar treatments resulted in a higher CEC compared to other treatments (Figure 2b).

The highest available soil nutrients were recorded during the T_10_ treatment in both seasons, while the lowest was recorded during the T_1_ treatment. Specifically, T_10_ increased the soil’s available N in both seasons by 130.12% and 161.45%, available P by 89.49% and 102.02%, and available K by 39.84% and 70.45%, respectively, compared to the control treatment (Figure 3). Insignificant differences were found in soil available nutrients between all co-application treatments in both seasons. Additionally, the modified biochar increased the soil’s available nutrients more than the normal biochar.

### 3.2. Wheat Yields

Table 4 exhibits wheat yields. With the use of phosphogypsum and biochar treatments, wheat grain and straw yields increased significantly.

The highest grain and straw yields of wheat plants were achieved with the T_10_ treatment (2.72 and 2.92 Mg/ha for grain yield and 4.88 and 4.91 Mg/ha for straw yield) in the 2021/2022 and 2022/2023 seasons, respectively (Figure 4). The mixed application of T_10_ resulted in a 41.36% and 58.55% increase in grain yield and a 52.69% and 59.32% increase in straw yield in both seasons, respectively, compared to the T_1_ treatment.

### 3.3. The Correlation Analyses

The PCA biplot analysis in Figure 5 provides valuable insights into the relationships between various soil parameters, wheat yield, and the different treatments. With two dimensions explaining 92.9% of the total variance, it is evident that these dimensions capture the majority of the variability in the dataset.

Dim1, which accounts for 83.8% of the variance, highlights a positive connection between soil organic carbon (SOC), cation exchange capacity (CEC), available nitrogen, and available potassium. This suggests that these factors are closely related and may collectively influence soil quality and fertility.

On the other hand, Dim2, explaining 9.1% of the variance, reveals a positive correlation between available phosphorus and wheat grain and straw yield. This dimension emphasizes the importance of phosphorus availability in influencing wheat productivity.

Moreover, the distinct separation of treatments along Dim1 and Dim2 underscores the differential effects of the various treatments on soil parameters and wheat yield. Specifically, the combined use of phosphogypsum and modified cotton stalk biochar (T_10_) appears to have the most significant impact, as indicated by its position on the PCA plot.

Interestingly, the correlation analyses suggest that improvements in soil characteristics, such as SOC, CEC, and nutrient availability, have a greater influence on wheat yield. This underscores the importance of enhancing soil quality to optimize crop productivity. Overall, these findings provide valuable insights for optimizing agricultural practices to improve soil health and maximize crop yields.

## 4. Discussion

The organic carbon content in soil (SOC) serves as an indicator of soil health and function. Significant improvements in SOC, CEC, soil nutrient content, and wheat grain and straw yield were obtained during the mixed application of phosphogypsum and modified biochar. This combined application strategy appears to have a synergistic effect, leading to substantial improvements across multiple soil parameters and crop productivity metrics.

By integrating phosphogypsum, which provides essential nutrients and amendments, with modified biochar, which enhances soil structure, nutrient retention, and microbial activity, a comprehensive approach to soil management was achieved. The observed increases in SOC indicate improved soil organic matter levels, which are vital for soil fertility, water retention, and overall soil health. Additionally, the enhancements in CEC suggest an improved capacity for nutrient retention and exchange within the soil, facilitating better nutrient availability to plants. Some studies have indicated that soil organic matter values increase with the addition of phosphogypsum [9,15,61]. The increase in SOM and CEC in sandy soil could be through the root residues in-depth [17,18,19,20,21]. The application of phosphogypsum results in notable changes in crop yields, plant nutrition, and soil fertility [11,21,62,63].

Our results indicated that the utilization of biochar treatments had a more pronounced impact on SOM compared to phosphogypsum (Figure 1). This can be attributed to the inherent carbon-rich nature of biochar, which serves as a stable reservoir of organic carbon in the soil. As biochar decomposes slowly over time, it releases organic carbon into the soil, thereby enriching SOM levels and contributing to soil fertility and health. These results align with findings from previous studies [23,24,25,26,41]. The organic carbon and nitrogen compounds present in biochar play a crucial role in mitigating soil carbon respiration loss and enhancing soil fertility [64,65]. Incorporating biochar into sandy soil significantly increases soil organic matter (SOM) [66,67]. By serving as a substrate, it gives microorganisms a place to live, which increases their activity and speeds up the breakdown of organic matter [68]. As a result, this encourages microbial populations, which increases the mineralization of soil carbon [69]. El-Naggar [31] noted that the addition of biochar improved the CEC and soil fertility in sandy loam soil. Furthermore, it has been shown that adding biochar to soils greatly raises the CEC of the altered soils [70,71]. Biochar lowers the possibility of nutrient loss from the soil system by facilitating increased nutrient availability for plants [72,73]. Increases in soil organic carbon, total nitrogen (N), and phosphorus (P) have been reported with the use of biochar. As such, even in the long run, its application remains a viable method for increasing agricultural productivity. This efficacy is associated with enhanced nutrient utilization efficiency in addition to changes in soil structure and carbon stocks [42]. Research has demonstrated that applying biochar can increase the availability of nutrients in the soil [33,74,75,76]. However, Hussain et al. [38] reported that excessive biochar application in alkaline soil conditions led to decreased maize and wheat yields due to nitrogen and micronutrient immobilization, making it less suitable for plant growth. Furthermore, the pH values of biochar are generally high, frequently higher than 8.0, indicating that it is an alkaline substance [77,78,79]. 

As a result, different biochar modification protocols, such as thermal, chemical, or physical treatments, have drawn interest [44,45,46,47,48,49,50]. Also, our findings were corroborated by the data; modified biochar has been found to absorb nutrients from the soil more effectively than unmodified biochar. This heightened nutrient absorption capability can be attributed to two key factors: the high-cation-exchange capacity (CEC) of modified biochar and its low-pH value (Table 2). These findings are consistent with previous research, supporting the notion of Sahin et al. [80] that modifying biochar with strong acids could potentially enhance nutrient availability in soil. Such increases in nutrient availability might contribute to enhanced soil fertility and improved plant nutrition. Additionally, modified biochar has demonstrated a greater increase in CEC than regular biochar, which significantly reduces nutrient leaching from sandy loam soil and increases retention, potentially enhancing plant nutrient uptake [48]. Although modified biochar has shown greater phosphorus (P) availability, its consistency varies at different incubation times [43]. However, the half-life of the available P in modified biochar exceeded 80% compared to control soil, indicating its potential as a slow-release fertilizer [53].

Furthermore, the positive impact on wheat grain and straw yields underscores the effectiveness of the combined application of phosphogypsum and modified biochar in promoting crop growth and productivity. These findings highlight the potential of integrated soil amendment strategies involving phosphogypsum and modified biochar to enhance soil quality, nutrient cycling, and agricultural sustainability. 

The utilization of phosphogypsum (PG) has been linked to enhanced wheat yields, as evidenced by studies conducted by [10,11,55]. Furthermore, da Costa et al. [17] showed that adding phosphogypsum to soil improves its fertility, which can benefit wheat nutrition, grain yield, and grain quality, as well as contribute to the sustainability of agricultural systems, especially when the soil is used intensively. It is possible that improvements in soil fertility, which are associated with increased root development across the soil profile, are responsible for the observed rise in wheat grain and straw yields [20]. 

The application of biochar was demonstrated to enhance wheat grain and straw yields, a trend that aligns with findings reported in earlier studies Zhao et al. [28], which demonstrated that biochar-fertilized soils led to enhanced root growth, wheat production, and nutrient uptake. Zhang et al. [35] has consistently shown that applying biochar increases wheat yield. Ali et al. [30] observed significant influences on maize yields in sandy soil with biochar and phosphogypsum application compared to recommended nitrogen fertilizers. Moreover, modified biochar has been found to contribute to higher grain and straw yields for wheat plants compared to unmodified biochar. Similarly, research by El-Sharkawy [50] has shown that applying modified biochar increases wheat yield more than normal biochar.

Optimizing soil quality is crucial for maximizing crop productivity and ensuring sustainable agricultural practices. The correlation analyses indicating a strong relationship between soil characteristics like soil organic carbon (SOC), cation exchange capacity (CEC), nutrient availability, and wheat yield highlight the importance of soil health management in agricultural systems. Furthermore, understanding the specific impacts of different treatments, as elucidated by the PCA biplot analysis, allows for targeted interventions to optimize agricultural practices. For instance, the identification of the combined use of phosphogypsum and modified cotton stalk biochar (T_10_) had the most significant impact; this suggests that implementing such treatments could be particularly beneficial for improving soil parameters and enhancing wheat yields.

## 5. Conclusions

Certainly, the amalgamation of phosphogypsum and biochar derived from agricultural waste, such as rice straw and cotton stalks, is proven to be an optimal and sustainable approach for repurposing substantial quantities of phosphogypsum and agricultural waste. This strategy holds promise for enhancing the quality of sandy soil and promoting crop production. Modified biochar presents promising outcomes in enhancing soil properties through improved porous functions, surface functional groups, and mineral compositions. Phosphogypsum and cotton-stalk-modified biochar treatment demonstrated comparable efficacy in increasing SOC, CEC, and nutrient availability in soil and resulted in remarkable wheat grain yield increases of 41.36% and 58.55%, along with straw yield increases of 52.69% and 59.32%. Overall, the findings suggest that mixed phosphogypsum and cotton-stalks-modified biochar treatment hold promise for improving loamy sand soil quality and wheat productivity. Furthermore, this practice adheres to sustainability principles by repurposing waste materials in a way that enhances agricultural ecosystems and reduces environmental impact.

## Figures and Tables

**Figure 1 plants-13-01492-f001:**
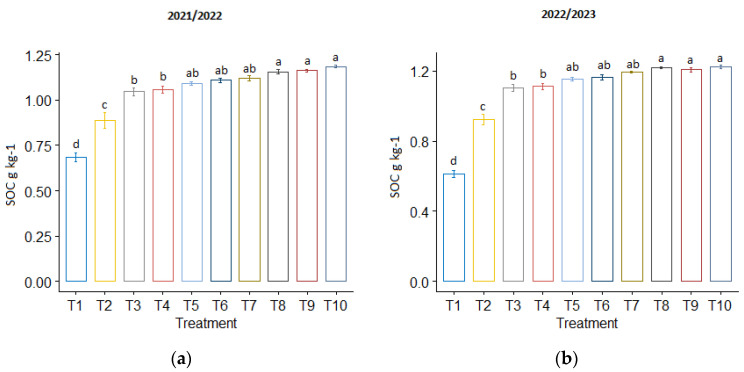
The influence of phosphogypsum and biochar treatments on soil organic carbon (SOC) following wheat harvest during (**a**) the first season of 2021/2022 and (**b**) the second season of 2022/2023. The standard error of the mean (SE) is shown at the top of the bars.

**Figure 2 plants-13-01492-f002:**
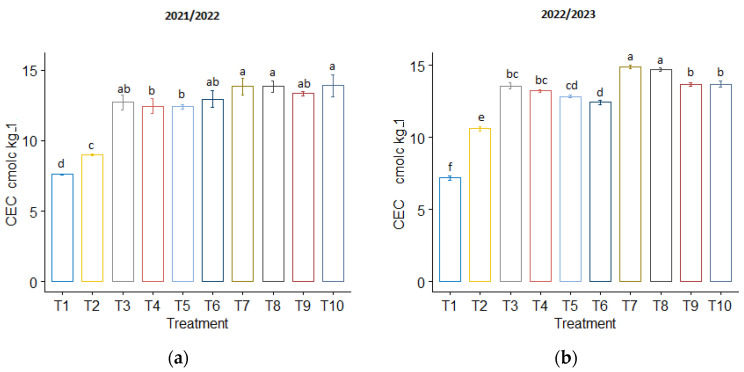
The influence of phosphogypsum and biochar treatments on cation exchange capacity (CEC) following wheat harvest during (**a**) the first season of 2021/2022 and (**b**) the second season of 2022/2023. The standard error of the mean (SE) is shown at the top of the bars.

**Figure 3 plants-13-01492-f003:**
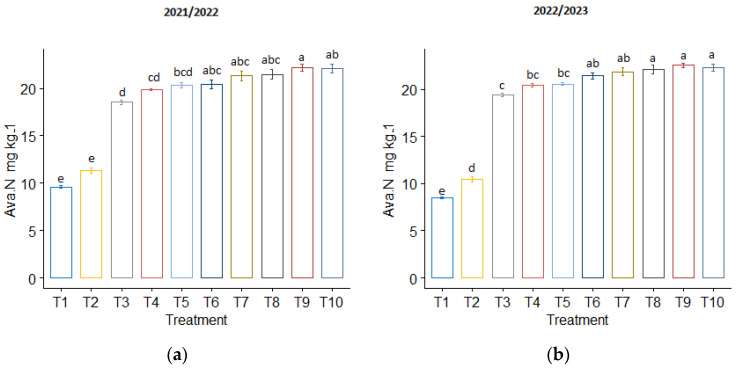
The influence of phosphogypsum and biochar treatments on available soil nutrients following wheat harvest: (**a**) Available nitrogen in 2021/2022, (**b**) Available nitrogen in 2022/2023, (**c**) Available phosphorus in 2021/2022, (**d**) Available phosphorus in 2022/2023, (**e**) Available potassium in 2021/2022, and (**f**) Available potassium for 2022/2023. The standard error of the mean (SE) is shown at the top of the bars.

**Figure 4 plants-13-01492-f004:**
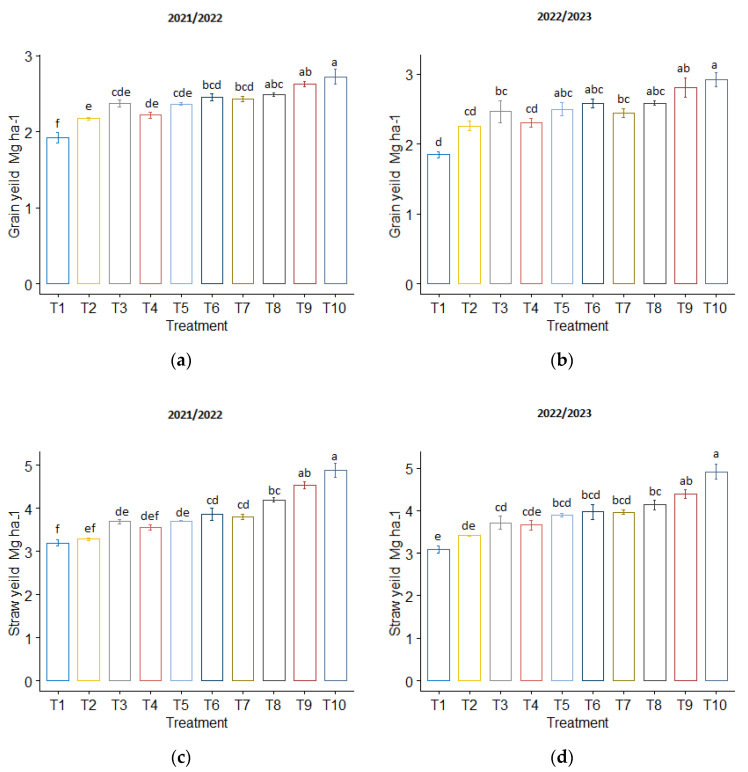
The influence of phosphogypsum and biochar treatments on wheat grain yield (Mg ha^−1^) (**a**) for the year 2021/2022 and (**b**) for the year 2022/2023; and straw yield (Mg ha^−1^): (**c**) for the year 2021/2022 and (**d**) for the year 2022/2023. The standard error of the mean (SE) is shown at the top of the bars.

**Figure 5 plants-13-01492-f005:**
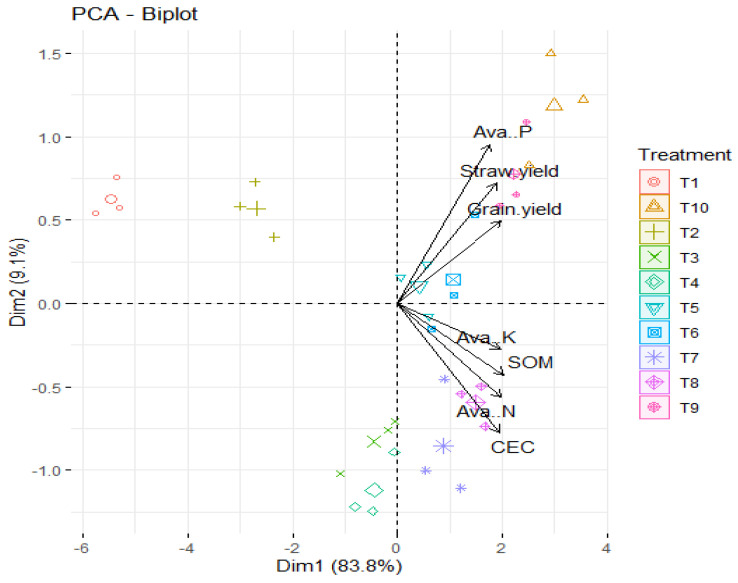
The relationships between various soil parameters and wheat yield during different treatments.

**Table 1 plants-13-01492-t001:** The soil characteristics and nutrient levels before experiment setup.

Parameters		Value
Particle size distribution (%)	Sand	64.21
	Silt	7.56
	Clay	28.23
The soil texture	loamy sand	
The soil electrical conductivity (EC, dS m⁻^1^)		3.56
Soil pH		7.56
Bulk density (Mg/m^3^)		1.52
Soil water content (%)		27.48
Soil organic carbon content (g/kg)		0.709
Cation exchange capacity (CEC, cmol_c_/kg)		7.51
Available N (mg/kg)		9.29
Available P (mg/kg)		6.45
Available K (mg/kg)		79.13

**Table 2 plants-13-01492-t002:** Chemical characterization of raw materials, biochar (B), and modified biochar (MB).

Parameters	Rice Straw		Cotton Stalks	
	RB	RMB	CB	CMB
pH	7.30	5.51	7.51	5.38
EC (dS m^−1^)	1.43	0.89	1.56	1.23
C%	63.8	48.1	76.9	61.7
N%	1.54	1.38	2.08	1.92
P%	0.557	0.438	0.634	0.616
K%	1.32	1.13	6.78	4.02
CEC (cmol^+^ kg^−1^)	37.61	55.93	41.77	60.95

**Table 3 plants-13-01492-t003:** The influence of phosphogypsum and biochar treatments on soil organic carbon (SOC), cation exchange capacity (CEC), and available soil nutrients following wheat harvest in 2021/2022 and 2022/2023.

Seasons			2021/2022				2022/2023			
Parameters		df	SS	MS	F Value	*p* Value	SS	MS	F Value	*p* Value
SOC (g kg^−1^)	Treatment	9	0.6448	0.07164	60.20	1.59 × 10^−12^ ***	0.9721	0.10802	145.968	3 × 10^−16^ ***
	Residual	20	0.0238	0.00119			0.0148	0.00074		
CEC (cmol_c_ kg^−1^)	Treatment	9	126.06	14.007	21.81	1.84 × 10^−8^ ***	139.75	15.528	231.72	<2 × 10^−16^ ***
	Residual	20	12.85	0.642			1.34	0.067		
N (mg kg^−1^)	Treatment	9	548.1	60.9	155.95	<2 × 10^−16^ ***	701.5	77.95	292.60	<2 × 10^−16^ ***
	Residual	20	7.8	0.39			5.3	0.27		
P (mg kg^−1^)	Treatment	9	104.39	11.598	86.76	4.77 × 10^−14^ ***	126.93	14.104	184.90	<2 × 10^−16^ ***
	Residual	20	2.67	0.134			1.53	0.076		
K (mg kg^−1^)	Treatment	9	2896.8	321.9	7.50	9.48 × 10^−5^ ***	6134	681.5	15.30	3.83 × 10^−7^ ***
	Residual	20	858.7	42.9			891	44.5		

‘***’ for *p* < 0.001, and ‘ ’ (empty) for *p* ≥ 0.05.

**Table 4 plants-13-01492-t004:** The influence of phosphogypsum and biochar treatments on wheat grain (G.Y, Mg ha^−1^) and straw yields (S.Y, Mg ha^−1^).

Seasons			2021/2022				2022/2023			
Parameters		df	SS	MS	F Value	*p* Value	SS	MS	F Value	*p* Value
G.Y (Mg ha^−1^)	Treatment	9	1.425	0.15833	21.50	2.08 × 10^−8^ ***	2.4198	0.26887	10.65	7.08 × 10^−6^ ***
	Residual	20	0.1473	0.00736			0.5049	0.02524		
S.Y (Mg ha^−1^)	Treatment	9	7.577	0.8419	40.32	6.99 × 10^−11^ ***	6.97	0.7745	19.34	5.26 × 10^−8^ ***
	Residual	20	0.418	0.0209			0.801	0.0401		

‘***’ for *p* < 0.001, and ‘ ’ (empty) for *p* ≥ 0.05.

## Data Availability

Data sets analyzed during the present study are accessible from the current author on reasonable request.

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
