# Peer review of "The Concurrent Application of Phosphogypsum and Modified Biochar as Soil Amendments Influence Sandy Soil Quality and Wheat Productivity"

_plants, 2024, doi:10.3390/plants13111492_

Round 1

Reviewer 1 Report

Comments and Suggestions for Authors

The manuscript entitled ‘’The concurrent application of phosphogypsum and modified biochar as soil amendments influences sandy soil quality and wheat productivity’’ investigated the impact of combining phosphogypsum and modified biochar as environmentally friendly soil amendments on loamy sand soil quality and wheat yield. The idea of the paper is interesting. However, there are some issues and should be modified especially in the introduction and discussion sections. Please see the comments as below:

Abstract:

Abstract provides a clear objective regarding the study and treatments. However, it would be beneficial to explicitly state the specific aims or hypotheses being tested.

Introduction:

Line 37: 96% of Egypt's total land area are covered with sandy soil?! Please check it!

The introduction needs more extension. What is previous research and what is the gap of knowledge? Compare them with your research and explain how the current research can filling this gap?

Also, I would suggest that the authors add some information about the different feedstock, pyrolysis temperatures and modified method on biochar characteristics in last paragraph. Here is a recently published that you can use it to improve the introduction: https://doi.org/10.1016/j.enconman.2023.117924

Results and Discussion:

Please add the axis title (vertical) for all the figures (figures 1-4).

Please check the significance letters! For example, Fig 1-a, significance letter in T3 is e, but for T4 is de!! Significant letter is base on the SD or SE and should be based on the software calculation!

Line 95: Figure 1 a or b?? all the Figures should be referred in the text. Please check for all the Figures.

Line 125: Table 3 -- > Table 2

The discussion is not well done. To interpretation of those you need to compare them with other studies. What is the strength or weakness of your results? What is your interpretation for them? Make your interpretation strong with other works.

Material and Methods:

4.1. Experimental location: Please add the latitude and longitude of the research.

It is better to bring the soil characteristics in the form of a table (lines 221-216).

Conclusion is well written.

References:

The authors used suitable references, so it clearly came from a good background and due to updated studies and related research.

Comments on the Quality of English Language

 Minor editing of English language required

Reviewer 2 Report

Comments and Suggestions for Authors

It is not clear why the author does not follow the standard organization of research papers. Commonly Materials and Methods are described before the Results. This should be changed.

Section 4.5, should be described in Materials and Methods which statistical tests were used.

It seems that in the experimental design, there are main effects as the application of phosphogypsum, the application of biochar, also nested effects as variants of biochar. As effects as untreated/modified biorar, rice/cotton biorach. It is not clear what model was used, maybe some type of linear model? Are there interactions or only the main effects are important? The description of statistical models and the results of these models should be more detailed. If the authors have not performed the analysis as the linear model, I advise the authors to perform such an analysis and present the influence of the main factors and influences. That may give some more insight into the studied phenomena than just performing ANOVA for 10 treatments. I only speculate what was the statistical approach applied by the authors as it is not explained in detail in the manuscript.

It would be interesting to present some correlations between the soil characteristics and yield. The authors discuss it in some way in the manuscript, as they are indicators of soil health, so influence the yield as one can assume. But no results demonstrate a level of such influence found in their experiment. 

In all figures legend is not needed as the same information is available in the axis, the legend is even confusing.

Table 1 is not clear, there are two sets of columns for parameters. I assume that they are for two studied periods. One can only guess that 2021/2022 are on the left, but it is not sure.

I'm wondering about the validity of the measurements in the experiment. In the figures, the authors present the whiskers at the bars, which I guess (the authors do not describe their meaning) represent the standard deviation of the measurements (or maybe the range?). The variability of the measurements of various samples is in most cases very small. The samples are rarely so similar. Maybe the authors should double-check these results and describe in more detail the methodology. It just seems uncommon and may be a significance of some errors in the experiment or data analysis.

Such an extremely small variability in data is presented even for yield. When there are 3 replications of size 2 m2 it seems very unlikely that the wheat grain from such a surface is the same for 3 replications (from the size of the bars it seems that in some cases the difference is at most a few % or even smaller). The same is true for straw yield and grain yield.

What is also strange, as a control treatment the authors chose soil treated with recommended NPK fertilizer. All other treatments with the addition of phosphogypsum or various biochar variants give significantly higher yields. So one can conclude, that phosphogypsum alone is much better than the recommended fertilizer or any other proposed treatment is also much better. Such surprising results need solid evidence. Maybe there is some bias in the choice of the control treatment, which was accidentally (or deliberately) chosen in such a way, that the comparison is unjust. Or there is some other mistake in the design of the experiment. The authors should think about that subject and discuss it in the paper.

What are the units in Fig. 1? In Table 1 the same quantity is presented and authors used their gram per gram. (g g-1). It seems strange that in this figure the results are above one. Even close to one seems to be a mistake.

Maybe I don't understand that, but what is the reason for the presentation of Table 1? That means that there are statistically significant differences between the treatments, but still, it is not clear between which pairs. More meanings have figures, but as I mentioned, I have doubts about the reported variability between the replications of the same treatment.

Maybe such a table should be moved to supplementary materials or an appendix.

Round 2

Reviewer 2 Report

Comments and Suggestions for Authors

I still insist on changing the order of the presented sections.

In the instructions to the authors of the Plants journal, the required sections are listed in more natural order.

"We do not have strict formatting requirements, but all manuscripts must contain the required sections: Author Information, Abstract, Keywords, Introduction, Materials & Methods, Results, Conclusions, Figures and Tables with Captions, Funding Information, Author Contributions, Conflict of Interest and other Ethics Statements."

The common practice is to have Materials & Methods at the beginning of the manuscript, which facilitates understanding.

The authors responded to my remark: "Response 9: Chemical fertilizer was applied in all treatments, except for those treated with

phosphogypsum, where phosphorus fertilizer was omitted. Therefore, I believe there are no

errors in selecting the control option."

That information should be written in the manuscript. Reading section 4.3 it seems that it was applied only to the control.

"Response 8 In the experiment, ten readings were initially taken. Subsequently, only three

replicates exhibiting closely aligned results were utilized."

That also should be specified in the text.
